# Acute Low-Dose Hyperoxia during a Single Bout of High-Intensity Interval Exercise Does Not Affect Red Blood Cell Deformability and Muscle Oxygenation in Trained Men—A Randomized Crossover Study

**DOI:** 10.3390/sports8010004

**Published:** 2020-01-04

**Authors:** Nils Freitag, Tim Böttrich, Pia D. Weber, Giorgio Manferdelli, Daniel A. Bizjak, Marijke Grau, Tanja C. Sanders, Wilhelm Bloch, Moritz Schumann

**Affiliations:** 1Department of Molecular and Cellular Sport Medicine, Institute of Cardiovascular Research and Sport Medicine, German Sport University Cologne, Am Sportpark Müngersdorf 6, 50933 Cologne, Germany; n.freitag@dshs-koeln.de (N.F.); tim@boettrich.net (T.B.); pia.weber@dshs-koeln.de (P.D.W.); d.bizjak@dshs-koeln.de (D.A.B.); m.grau@dshs-koeln.de (M.G.); w.bloch@dshs-koeln.de (W.B.); 2Department of Biomedical Sciences for Health, School of Exercise Sciences, Università degli Studi di Milano, 20122 Milan, Italy; gmanferdelli@gmail.com; 3Department of Preventive and Rehabilitative Sport Medicine, Institute of Cardiovascular Research and Sport Medicine, German Sport University Cologne, Am Sportpark Müngersdorf 6, 50933 Cologne, Germany; t.sanders@dshs-koeln.de

**Keywords:** oxygen therapy, oxygen supplementation, FiO_2_, oxygen kinetics, hemorheologic response, NIRS, interval exercise

## Abstract

Recent technological developments provide easy access to use an artificial oxygen supply (hyperoxia) during exercise training. The aim of this study was to assess the efficacy of a commercially available oxygen compressor inducing low-dose hyperoxia, on limiting factors of endurance performance. Thirteen active men (age 24 ± 3 years) performed a high-intensity interval exercise (HIIE) session (5 × 3 min at 80% of W_max_, separated by 2 min at 40% W_max_) on a cycle ergometer, both in hyperoxia (4 L∙min^−1^, 94% O_2_, HYP) or ambient conditions (21% O_2_, NORM) in randomized order. The primary outcome was defined as red blood cell deformability (RBC-D), while our secondary interest included changes in muscle oxygenation. RBC-D was expressed by the ratio of shear stress at half-maximal deformation (SS^1^/_2_) and maximal deformability (EI_max_) and muscle oxygenation of the rectus femoris muscle was assessed by near-infrared spectroscopy. No statistically significant changes occurred in SS^1^/_2_ and EI_max_ in either condition. The ratio of SS^1^/_2_ to EI_max_ statistically decreased in NORM (*p* < 0.01; Δ: −0.10; 95%CI: −0.22, 0.02) but not HYP (*p* > 0.05; Δ: −0.16; 95%CI: −0.23, −0.08). Muscle oxygenation remained unchanged. This study showed that low-dose hyperoxia during HIIE using a commercially available device with a flow rate of only 4 L·min^−1^ may not be sufficient to induce acute ergogenic effects compared to normoxic conditions.

## 1. Introduction

Oxygen uptake, transport, and use are crucial domains of endurance capacity [1,2]. Several determinants, such as pulmonary diffusion and cardiac output but also peripheral factors such as the oxygen carrying capacity of the blood and mitochondrial density of skeletal muscle are affecting the oxygen flux [1]. While pulmonary diffusion, cardiac output and mitochondrial density may adapt during chronic training, mainly the oxygen carrying capacity can be influenced acutely [1] and, thus, might be of interest for short-term improvements in endurance capacity.

The majority of methods to manipulate the carrying capacity are banned by the World Anti-Doping Agency (WADA) [3]. However, supplemental oxygen (hyperoxia) is currently not restricted by the WADA and has the potential to affect these contributors as it was shown to increase arterial oxygen pressure and, thus, causes higher arterial oxygen saturation and the freely dissolved oxygen fraction in the blood [2,4,5,6]. Thus, hyperoxia was previously shown to improve acute endurance performance during cycling [7,8], rowing [9,10], swimming [11] and running [12], as represented by increased rates of maximal power output and peak oxygen consumption (VO_2peak_).

The potential theories behind the acute beneficial effects of hyperoxia on endurance performance were summarized in several systematic reviews [2,5,6]. It appears that enhanced performance is likely to be induced by a maintained muscle [5] and/or cerebral oxygenation [10,13]. This in turn would induce an improved rate of oxygen diffusion [6], possibly leading to an improved neural drive [14]. Furthermore, an increased oxidative phosphorylation with simultaneously decreased use of phosphocreatine as well as lowered lactate accumulation were reported [15,16,17,18,19] and some evidence also emerged that an increased inspired fraction of oxygen (FiO_2_) might counteract exercise-induced arterial hypoxemia (EIAH), which is thought to have detrimental effects on aerobic performance [20,21].

Maintaining systemic oxygen saturation as well as oxygen supply to the working muscle are considered to be key mechanisms of exercise performance but seem to be dependent on numerous physiological factors, including haemoglobin mass, blood viscosity and red blood cell deformability (RBC-D) [22,23,24]. In addition to the oxygen supply, the RBC-D and tissue oxygenation are affecting phosphocreatine resynthesizes during recovery, leading to a direct impact on exercise capacity [24,25,26]. A higher RBC-D might increase oxygen diffusion from the alveoli to pulmonary capillaries, resulting in higher blood oxygen content and, thus, in a greater muscle oxygenation and potentially decreased fatigue [27,28]. In contrast, a reduced tissue oxygen saturation is linked to alterations in blood viscosity, aggregation and deformability as well as an increased flow resistance, possibly leading to an inhibition of microcirculation and tissue oxygen perfusion [29]. Therefore, increasing the rate of inspired oxygen might counteract a reduced performance capacity by reducing the exercise-induced local or systemic hypoxia.

Recent technological developments provide easy access to use hyperoxia during training and recovery and, thus, its use in daily training practice is constantly increasing [2,5,30]. However, the availability of numerous devices also bears difficulties because the supplied oxygen concentrations are highly dependent on the technology used. Typically, oxygen concentrations are ranging from 30% to 100% with flow rates of 2 to 20 L per minute [2,5,31]. The rate of flow is depending on whether compressors or gas cylinders are used and whether the oxygen is supplied through nasal cannulas, rebreather masks or oxygen on demand systems [30]. Furthermore, it needs to be considered whether the oxygen is humidified or dry, liquid or gaseous and whether oxygen is supplied in normobaric or hyperbaric conditions [31,32,33,34,35]. Lindholm and colleagues pointed out that oxygen compressors are more practical compared to gas tanks, depending on training duration, frequency, and intensity. Furthermore, oxygen compressors are favourable in terms of storage and safety. However, oxygen compressors are converting surrounding air into concentrated oxygen, which causes a lower flow rate compared to tanks with liquid or gaseous oxygen. [30].

A recent meta-analysis indicated a moderate-positive correlation between FiO_2_ and acutely enhanced endurance performance, whereby largest improvements in performance (i.e., time-trial, time to exhaustion or graded exercise tests) were observed with a FiO_2_ of ≥ 0.3 [5]. However, even a FiO_2_ of 0.26 was previously shown to be sufficient to maintain oxygen saturation and counterbalance EIAH both in trained and highly trained athletes [36]. Considering the growing interest of coaches and athletes alike as well as an easier access to oxygen-delivering devices, the rationale of this study was to assess the efficacy of a commercially available oxygen compressor with a maximal flow rate of 4 L min^−1^ (HYPOXcontrol, Medicap Homecare GmbH, Ulrichstein, Germany).

The primary outcome of interest was the supplemental oxygen-induced change in RBC-D. Secondary interests included changes in muscle oxygenation, blood lactate concentrations (BLa), heart rate (HR) and ratings of perceived exertion (RPE). Considering the previous reported positive effects of acute hyperoxia on physiological function with low-dose hyperoxia [36], we followed the following hypothesis for our primary outcome of interest: (i) a flow rate of 4 L∙min^−1^ will improve RBC-D to a greater extent than normoxia; (ii) low-dose hyperoxia will be sufficient to maintain muscle oxygenation while it will reduce RPE, HR and BLa concentrations in physically active subjects performing high-intensity interval exercise.

## 2. Materials and Methods

### 2.1. Participants and Ethical Aspects

Thirteen men participated in this randomized crossover trial (Table 1). The participants were all sport science students fulfilling recommendations for habitual exercise, as outlined by the American College of Sports Medicine [37] but did not follow a structured endurance exercise program. Participants were informed about possible risks of all study procedures before giving a written informed consent. All subjects were non-smokers and abstained from alcohol, caffeine, and vigorous activity the day before the trial. A cardiologist reviewed a completed health questionnaire and resting electrocardiogram prior to the first testing. Subjects were free of acute and chronic illness, disease or injury and did not report intake of medication that would contraindicate the performance of intense exercise. The study was conducted according to the Declaration of Helsinki. The Ethics Committee at the German Sport University (004/2017) granted ethical approval.

### 2.2. Study Design

This was a single-blind randomized crossover study (Figure 1). An independent technician not involved in any study procedures conducted the sequence randomization using the PC software RITA (Randomization In Treatment Arms, Evident, Lübeck, Germany). After pre-screening and initial aerobic performance testing, all participants performed a high-intensity interval exercise (HIIE) session in normoxia (NORM) and hyperoxia (HYP) in randomized and counterbalanced order, each separated by at least 48 h. Due to logistical reasons the participants and supervising exercise personnel were not blinded to the condition; however, the investigator conducting the outcome analysis was not aware of the sequence order of the participants. The exercise intensities (80% and 40% W_max_) were determined by percentage of the maximal power output (W_max_) obtained during an initial incremental cardiopulmonary exercise testing on a cycle ergometer in normoxic condition. Capillary blood samples were drawn from the earlobe both before (Pre) and immediately after (Post) each exercise session to assess RBC-D, and before and after each high-intensity bout to assess BLa concentrations. Similarly, venous blood samples were collected prior to and immediately after each session in order to assess a basic blood count. HR, RPE and near-infrared spectroscopy (NIRS) were monitored continuously throughout each exercise bout. To standardize the measurement conditions, participants were asked to maintain similar nutritional intake prior to each exercise session.

### 2.3. Aerobic Performance

Peak oxygen uptake (VO_2peak_) of the subjects was determined by an incremental exercise test to exhaustion on a bike ergometer (Excalibur Sport, Lode BV, Groningen, the Netherlands) in ambient air. All incremental exercise tests were performed in the morning between 7 a.m. and 9 a.m. The protocol commenced with an initial load of 30 W and increased by 40 W every 3 min [38]. Participants were asked to maintain a pedalling frequency between 65 and 70 revolutions per minute throughout the test. The test was terminated when participants failed to keep up the required frequency for more than 15 s, despite verbal encouragement. HR was monitored throughout the test (Polar A300; Polar Electro Oy, Kempele, Finland) and reported as the average of the final 15 s of each increment, together with RPE. Oxygen uptake was determined continuously breath-by-breath using a gas analyser (Zan600, nSpire Health GmbH, Oberthulba, Germany). On each testing day, gas calibration with reference gas as well as airflow calibration was performed using a manual flow calibrator. VO_2peak_ used for statistical analysis was calculated as the highest VO_2_ value averaged over 30 s. In addition, W_max_ was calculated using the following equation:*W_max_* = *W_com_* + 40 (*t* ∙ 180^−1^)(1)
with *W_com_* defined as the wattage of the last completed increment with *t* as the time of the last incomplete increment (in seconds) [39].

### 2.4. Exercise Loading

Each protocol consisted of five high-intensity bouts of 3-min at 80% of W_max_, separated by 2-min at 40% of W_max_. Participants were instructed to maintain a constant pedalling frequency between 65 and 70 revolutions per minute during each exercise session. HYP was supplied by a commercially available oxygen compressor with a flow rate of 4 L·min^−1^ with 94% O_2_ (HYPOXcontrol, Medicap Homecare GmbH, Ulrichstein, Germany). The gas mixture was delivered through a full-faced respiration mask, where it was further mixed with ambient air. NORM was performed at ambient room conditions. BLa, HR and RPE (BORG scale [40]) were assessed before and after each exercise loading, as well as before and after each high-intensity bout. The protocol was modified from the Norwegian 4 × (4 + 3 min) high-intensity protocol as it was proven to be heavily oxygen dependent [41,42,43]. However, we modified this protocol to 5 × (3 + 2 min) in order to make it comparable to other clinical studies of our lab [43,44].

### 2.5. Primary Outcome

#### Red blood cell deformability

It was previously shown that the sampling technique does not affect markers of RBC-D [45]. Therefore, capillary sampling was used for practical reasons. Capillary blood samples from the earlobe were anticoagulated using sodium heparin and immediately mixed with an isotonic viscous polyvinylpyrrolidone (PVP) solution (0.14 mM, osmolality 300 mOsm L^−1^, viscosity 29.80 mPa s at 37 °C, Mechatronics, the Netherlands) in a 1:250 ratio (vol:vol). RBC-D was measured with a laser-assisted optical rotational cell analyser (LORCA; RR Mechatronics, Hoorn, The Netherlands) as described elsewhere [46]. Briefly, the blood/PVP sample was sheared in a Couette system with nine consecutive shear stresses ranging from 0.3 to 50 Pa. Shearing of RBC resulted in a diffraction pattern of the laser beam that was directed through the sample. Diffraction pattern was analysed by the LORCA software and resulting elongation indices (EI) where used to calculate the maximum deformability (EI_max_), which is defined as the theoretical maximal deformability at infinite shear stress and SS^1^/_2_ (Pa), which can be defined as the shear stress at half-maximal deformation. The SS^1^/_2_ to EI_max_ ratio was then calculated because it was previously shown that the ratio is more robust in reflecting alterations in RBC-D compared to the single parameters presentation [46,47]. A lower ratio of SS^1^/_2_ to EI_max_ indicates higher RBC-D [47].

### 2.6. Secondary Outcomes of Interest

#### 2.6.1. Muscle Oxygenation

Near-infrared spectroscopy (NIRS, moorVMS-PC 4.1, Moor Instruments, Devon, UK) was used to assess tissue oxygen saturation (SO_2_; also known as tissue saturation index (TSI)), oxygenated haemoglobin (and myoglobin) concentration (oxyHb) and deoxygenated haemoglobin (and myoglobin) concentration (deoxyHb) during each HIIE protocol. In addition, total tissue haemoglobin (tHb) was calculated from the summation of oxyHb and deoxyHb concentrations. Alterations in tHb are known to be an indicator of changes in perfusion by reflecting adjustments in total tissue blood volume, while changes in SO_2_ provide information on the balance of O_2_ supply and consumption [48,49].

A pair of NIRS optodes was placed on the rectus femoris muscle of left leg at the half distance between the anterior superior iliac spine and the patella and fixed by double-sided adhesive tape. It was previously shown that the total muscle activation of the rectus femoris muscle and the vastus lateralis muscle was quite similar [50,51]. In addition, it was indicated that the ratio of the percentage changes of deoxyHb relative to the muscle activation did not differ between the rectus femoris and vastus lateralis muscles [52] and, thus, the rectus femoris was used for NIRS assessment to possibly further reduce potential sliding due to gravity when compared to placement on the vastus lateralis muscle.

A probe holder was used to assure a 30-millimetre separation between emitter and detector was used to protect the probes against external light sources and maintain the same inter-optode distance. The device was calibrated regularly based on manufacturer standards.

Signals were recorded continuously with a sampling rate of 40 Hz throughout the entire exercise session. The measures of oxyHb and deoxyHb were recorded in arbitrary units (AU), while SO_2_ was recorded as the percentage of oxyHb divided by tHb. The method is based on spatially resolved spectroscopy and modified for the law of Lambert–Beer [53].

#### 2.6.2. BLa, HR and RPE

Concentrations of capillary BLa (20 μL) were assessed at rest as well as before and after each high-intensity bout. The BLa samples were drawn from the earlobe and mixed with a haemolytic fluid. Thereafter, the samples were measured using a fully enzymatic amperometric instrument (EKF Biosen S-Line Analyser, Diagnostic GmbH, Barleben, Germany).

HR was recorded constantly throughout the test using a wrist-worn monitor coupled with a chest belt via Bluetooth (Polar A300; Polar Electro Oy, Kempele, Finland). The values of the final 15 s of each high-intensity bout were averaged and used for statistical analysis. Similarly, RPE was assessed using the BORG scale (6 to 20) [40] within the final 15 s of each interval.

#### 2.6.3. Basic Blood Count

Venous blood samples were drawn from the antecubital vein both before and immediately after each HIIE session into EDTA tubes (BD-Belliver Industrial Estate, Plymouth, UK). The basic blood count was determined with the KX−21N analyser (Sysmex Corporation, Kobe, Japan). Total red blood cell (RBC) count, haemoglobin (HGB) concentration and haematocrit (HCT) were determined.

### 2.7. Statistical Analysis

Statistical analyses were performed using IBM SPSS Statistics for Windows Version 26 (IBM Corp., Armonk, NY, USA). Data are expressed as mean ± SD, unless stated otherwise. Data were analysed using a repeated measure ANOVA with Bonferroni corrections. Greenhouse–Geisser correction was used if sphericity was violated. Effect sizes are presented as partial eta squared (η_p_^2^, ES). Statistical significance was accepted for all analyses at an alpha level of *p* < 0.05. Statistically significant trends were accepted for *p* < 0.1 and reported with exact values.

To assess the number of participants needed a calculation for sample size was performed to detect both statistically and clinically significant findings. The calculation for sample size was based on earlier studies investigating the effects of intense exercise on RBC-D, and was performed by G* power (version 3.1.9.2, Heinrich Heine University, Dusseldorf, Germany) [54], based on group x time interaction effects with a 2 × 2 repeated measures analysis of variance (repeated measures, within-between interactions). A total of 12 subjects was required to achieve an effect size of 0.8 with a power (1-*ß*) of 0.95. Based on our previous experience and the relatively short duration of the study, a drop-out rate of 10% was expected and consequently 13 subjects were included in the study.

## 3. Results

The maximal power achieved during the incremental exercise test was 310 ± 34 W and was obtained in 23:32 ± 02:05 min. The absolute workloads used during high-and low-intensity bouts were 259 ± 24 W and 122 ± 11 W, respectively. Analysis of the basic blood count revealed statistically significant main effects for time in RBC (F = 56.0, *p* < 0.001, η_p_^2^ = 0.7), HGB (F = 48.8, *p* < 0.001, η_p_^2^ = 0.7), and HCT (F = 55.9, *p* < 0.001, η_p_^2^ = 0.7) but not interaction (all *p* > 0.1) (Table 2).

### 3.1. RBC-D

In the ratio of SS^1^/_2_ to EI_max_ a statistically significant main effect was found for time (F = 15.2, *p* = 0.001, η_p_^2^ = 0.4) but not interaction (F = 0.7, *p* > 0.1, η_p_^2^ = 0.03). Further analysis revealed a statistically significant decrease from pre to post in NORM (*p* = 0.001) but not HYP (*p* > 0.1) (Figure 2).

### 3.2. Muscle Oxygenation

In oxyHb, no statistically significant main effects were found for time (F = 0.8, *p* > 0.1, η_p_^2^ = 0.03) or interaction (F = 0.7, *p* > 0.1, η_p_^2^ = 0.03) during high- and low-intensity bouts, respectively. Similarly, no statistically significant main effects were observed in deoxyHb for time (F = 0.5, *p* > 0.1, η_p_^2^ = 0.02) or interaction (F = 0.2, *p* > 0.1, η_p_^2^ = 0.009) (Figure 3). Also the analysis of tHb showed no statistical significant main effects for time (F = 1.0, *p* > 0.1, η_p_^2^ = 0.04) or interaction (F = 0.9, *p* > 0.1, η_p_^2^ = 0.04). Likewise, no statistically significant main effects were revealed in SO_2_ for time (F = 1.4, *p* > 0.1, η_p_^2^ = 0.06) or interaction (F = 0.1, *p* > 0.1, η_p_^2^ = 0.004).

### 3.3. BLa, HR and RPE

In BLa, a statistically significant main effect was found for time (F = 113.4, *p* < 0.001, η_p_^2^ = 0.8) but not interaction (F = 0.9, *p* > 0.1, η_p_^2^ = 0.04). Further analysis revealed a statistical increase from pre to post for both high- (HYP: *p* = 0.001; NORM: *p* < 0.001) and low-intensity (HYP: *p* < 0.001; NORM: *p* < 0.001) bouts (Table 2, Figure 4).

A statistically significant main effect in HR were observed for time (F = 286.4, *p* < 0.001, η_p_^2^ = 0.9) but not interaction (F = 1.4, *p* > 0.1, η_p_^2^ = 0.06). A significant increase over time was detected for high-(HYP: *p* < 0.001; NORM: *p* < 0.001) and low-intensity (HYP: *p* < 0.001; NORM: *p* < 0.001) intervals, respectively (Table 2, Figure 4).

In RPE, a statistical significant main effect for time (F = 220.7, *p* < 0.001, η_p_^2^ = 0.9) but not interaction (F = 1.6, *p* > 0.1, η_p_^2^ = 0.08) was found. A statistically significant increase from pre to post was observed for high-(HYP: *p* < 0.001; NORM: *p* < 0.001) and low-intensity (HYP: *p* = 0.001; NORM: *p* = 0.001) loading (Table 2).

## 4. Discussion

This study aimed to investigate the effects of low-dose hyperoxia during a high-intensity interval exercise session on RBC-D, tissue oxygenation and BLa using a commercially available oxygen compressor. Despite the small but statistically significant increase in RBC-D in normoxic conditions, we showed that RBC-D remained physiologically unaltered when HIIE was performed both in hyperoxia and normoxia. Furthermore, tissue oxygenation was maintained in both low-dose hyperoxia and normoxia and no significant between-condition effects were observed for the changes in BLa, HR and RPE. Therefore, our initial hypotheses were rejected.

Red blood cells have various important functions during acute exercise, namely assuring the O_2_-transport from the lungs to the periphery and the removal of metabolites, such as lactate and CO_2_ [24]. Therefore, it is of utmost interest, to investigate the responses of erythrocytes to changes in FiO_2_ concentrations, especially in order to gain more insights into the oxygen delivery kinetics. A reduced deformability combined with a higher flow resistance may alter blood flow and, therefore, oxygen kinetics by increasing the viscosity of blood [55]. Furthermore, reductions in RBC-D are linked to impaired tissue perfusion, increased metabolic stress and consequently a limited performance capacity [24]. Consequently, an improved deformability might be necessary to maintain the oxygen perfusion rates within the working muscle.

In the present study we observed only a small but statistically significant improvement in RBC-D after a single HIIE session in ambient condition. While it remains to be discussed whether this increase was physiologically meaningful, previous studies often showed increased RBC-D after acute exercise [56,57]. In general, hemorheologic responses to exercise seem to be dependent on intensity, volume and type of exercise with HIIE inducing the highest rates of shear stress [58]. Our findings showed further that RBC-D was not altered in the hyperoxic condition. In fact, we anticipated that an increased oxygen fraction in the inspired air improves oxygen loading of the RBCs and, therefore, improves oxygen supply to the muscles. However, while oxygen loading of haemoglobin was previously suggested to affect RBC-D [59], little is known on the oxygen loading kinetics of haemoglobin during low-dose hyperoxia or oxygen extraction within the microcirculation and whether this is depending on oxygen supply in the ambient air.

A possible mechanism, which would aid explaining our findings, may be related to the generation of oxidative stress. While there is uncertainty if increased FiO_2_ during HIIE has any effects on the anaerobic metabolism including BLa accumulation, energy use or production of reactive oxygen species (ROS), an increased ROS production may impact hemograms and hemorheologic parameters, such as RBC-D [60,61]. Hyperoxia was suggested to induce ROS production [28,62,63] and, although not measured, it might be speculated that hyperoxia-induced ROS might account for decreased RBC-D. It was indicated that an imbalanced increase in ROS concentrations inhibits cytoskeletal remodelling, might interact with nitric oxide, damages DNA and has cytotoxic effects leading to fast aging and destruction of RBCs [60,64].

In line with the findings for RBC-D, we also did not observe between-condition differences in tissue oxygenation, even though oxyHb and deoxyHb were somewhat larger after the administration of low-dose hyperoxia. This trend was also observed in other markers of tissue oxygenation, such as tHb but not in SO_2_. While in a recent meta-analysis it was shown that hyperoxia may indeed improve arterial saturation of oxygen, the effect on muscle oxygenation was moderate only [5]. Moreover, it was previously shown that hyperoxia may rather increase cerebral oxygenation, leading to an improved cellular diffusion of oxygen and oxygen-availability within the central nervous system rather than the periphery [13]. Similarly, it has to be considered that NIRS might be not sensitive enough to detect alterations in SO_2_ during high-intensity interval exercise as the body counter regulates the blood flow under intense exercise [65,66], leading to difficulties to distinguish between myoglobin and haemoglobin oxygenation [65].

Similarly, no major differences were observed for BLa accumulation in either condition, even though the effect size analysis revealed somewhat lower concentrations after HIIE with low-dose hyperoxia. Interestingly, most previous studies showed statistically reduced BLa concentrations for strenuous exercise performed in hyperoxia [5,8,19,67], while the underlying physiological mechanisms remain to be investigated. Recent theories suggest that an increased availability of oxygen leads to a shift from glycogenolysis towards an increased fat oxidation and/or an enhanced lactate clearance rate [15,18,19]. The absence of a significant attenuation of BLa accumulation in the present hyperoxic condition may possibly be explained by the relatively low increase in FiO_2_ (4 L·min^−1^ 94% O_2_, i.e., FiO_2_ ~0.3) administered. In fact, most previous studies typically used a FiO_2_ ranging from 0.5 to 0.6 [5] but the technology used in our study did not allow for this.

When interpreting our findings, one should bear in mind a few limitations. First of all, the provided oxygen delivery with a flow rate of 4 L·min^−1^ and 94% oxygen might have been too low to induce hemorheological changes during an intermittent loading. This is because especially in BLa and oxygen saturation visible trends for improvements in hyperoxia were observed but did not reach statistical significance. Moreover, it has to be acknowledged that in the present study no exercise-induced arterial desaturation under normoxia was observed and, thus, longer intervals or even a time to exhaustion protocol may be used in future investigations. Therefore, our findings may not be generalized to continuous type of aerobic exercise.

## 5. Conclusions

The primary findings of this study were that low-dose hyperoxia with 4 L·min^−1^ and 94% oxygen during HIIE had no effects on RBC-D and tissue oxygenation compared to HIIE in ambient conditions. As such, our findings imply that the use of such devices cannot be recommended to enhance acute athletic performance. However, one should bear in mind that this study examined only one device with a relatively small increase in FiO_2_ and, thus, future research should aim at comparing different oxygen concentrations.

## Figures and Tables

**Figure 1 sports-08-00004-f001:**
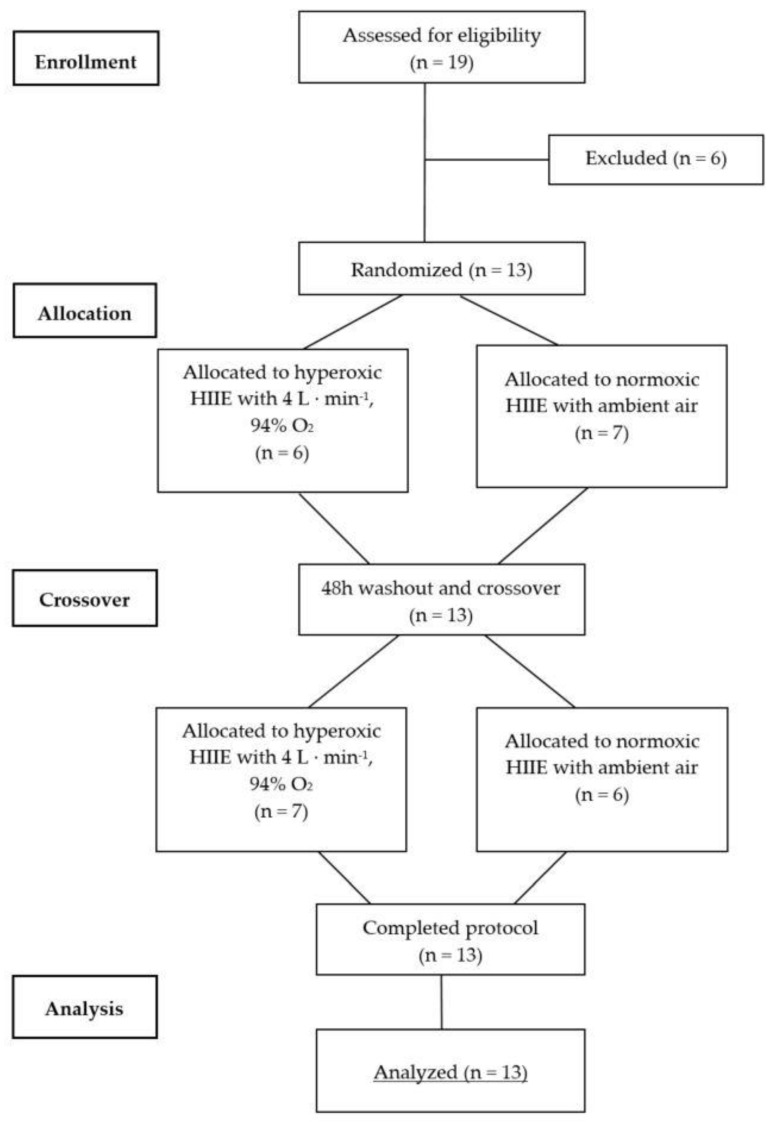
CONSORT flow diagram for randomized crossover trials. CONSORT = Consolidated Standards of Reporting Trials.

**Figure 2 sports-08-00004-f002:**
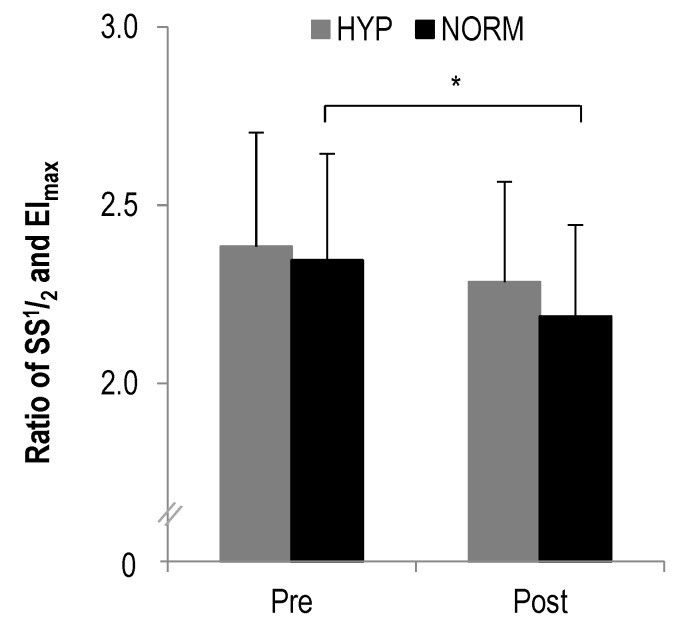
The ratio of SS^1^/_2_ to EI_max_ before and after each intervention, * statistically significant different to pre-test values (*p* < 0.01); values presented as mean ± SD.

**Figure 3 sports-08-00004-f003:**
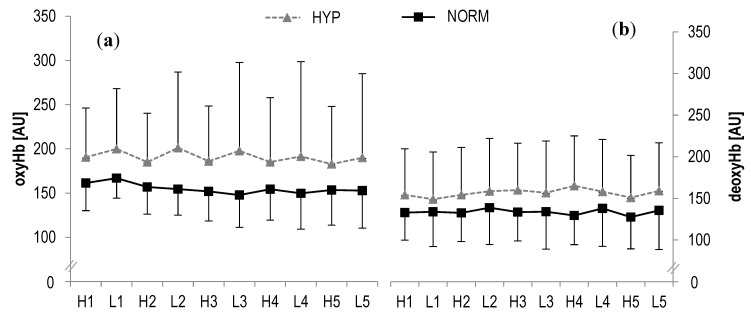
(**a**) Oxygenated haemoglobin concentrations (**oxyHb**) and (**b**) deoxygenated haemoglobin concentrations (deoxyHb) of the right rectus femoris muscle during HIIE for hyperoxia and normoxia; *H1-H5* high-intensity intervals number one to five; *L1-L5* active rest; values presented as mean ± SD.

**Figure 4 sports-08-00004-f004:**
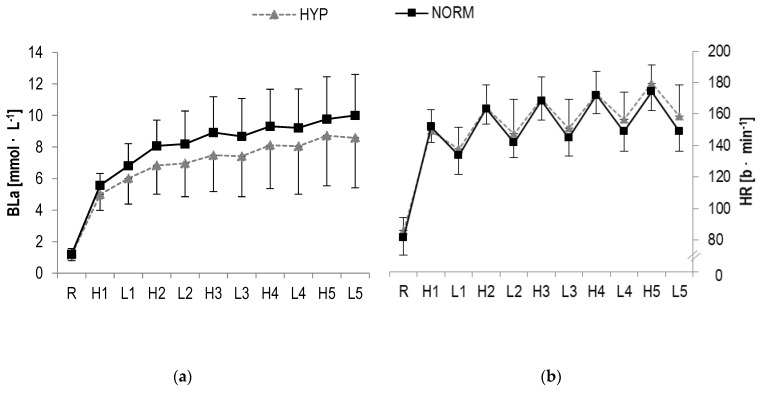
(**a**) Blood lactate concentrations (BLa), (**b**) Heart rate (HR) responses and (**c**) RPE during HIIE in hyperoxia and normoxia, respectively; *R* rest; *H1-H5* high-intensity intervals number one to five; *L1-L5* active rest; values presented as mean ± SD.

**Table 1 sports-08-00004-t001:** Subjects characteristics.

Characteristics	Values
N	13
Age [y]	23.6 ± 2.5
Height [cm]	184.8 ± 5.3
Weight [kg]	78.5 ± 9.0
Fat mass [kg]	11.2 ± 2.4
Fat mass [%]	14.1 ± 1.9
Lean mass [kg]	67.4 ± 7.0
Lean mass [%]	85.9 ± 1.9
VO_2peak_ [mL min^−1^∙kg^−1^]	53.3 ± 5.2
W_max_ [W]	310 ± 34
PO [W∙kg^−1^]	4.0 ± 0.2

N sample size; PO relative power output; VO_2peak_ peak oxygen uptake; W_max_ maximal power output; values presented as mean ± standard deviation.

**Table 2 sports-08-00004-t002:** Comparison of hyperoxia (HYP) and normoxia (NORM) from pre to post HIIE.

	HYP				NORM			
Parameters	Pre	Post	Δ	95%CI	Pre	Post	Δ	95%CI
RBC [10^6^ μL]	4.9 ± 0.3	5.2 ± 0.3 *	0.26	0.15; 0.37	5.0 ± 0.4	5.2 ± 0.5 *	0.23	0.14; 0.32
HGB [g dL^−1^]	15.4 ± 1.2	16.2 ± 1.1 *	0.80	0.44; 1.16	15.5 ± 1.3	16.2 ± 1.3 *	0.72	0.41; 1.03
HCT [%]	45.0 ± 2.9	47.3 ± 2.6 *	2.37	1.35; 3.39	45.4 ± 3.3	47.7 ± 3.6 *	2.27	1.38; 3.16
Ratio SS^1^/_2_ to EI_max_	2.38 ± 0.3	2.29 ± 0.3	−0.10	−0.22; 0.02	2.34 ± 0.3	2.19 ± 0.3 *	−0.16	−0.23; −0.08
BLa [mmol∙L^−1^]	1.2 ± 0.4	8.6 ± 3.0 *	7.4	5.42; 9.42	1.2 ± 0.3	10.0 ± 2.5 *	8.8	7.15; 10.45
HR [beats∙min^−1^ ]	85.9 ± 8.0	179.5 ± 11.2 *	93.6	84.4; 102.8	81.6 ± 10.6	174.3 ± 11.4 *	92.8	83.0; 102.6
RPE [6,7,8,9,10,11,12,13,14,15,16,17,18,19,20]	6.2 ± 0.4	19.1 ± 0.8 *	12.9	12.1; 13.7	6.1 ± 0.3	18.1 ± 1.5 *	12.0	10.8; 13.2

*BLa *blood lactate concentrations; HR heart rate; RPE ratings of perceived exertion; *WBC* white blood cells; *RBC* red blood cells; *HGB* haemoglobin; *HCT* haematocrit; *PLT* platelets; *^Δ^* absolute change from pre to post; *95% CI* lower and upper confidence intervals; *** statistically significant different to pre-test value (*p* < 0.001); values presented as mean ± SD.

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
