# Peer review of "Acute Low-Dose Hyperoxia during a Single Bout of High-Intensity Interval Exercise Does Not Affect Red Blood Cell Deformability and Muscle Oxygenation in Trained Men—A Randomized Crossover Study"

_sports, 2020, doi:10.3390/sports8010004_

Round 1
Reviewer 1 Report
Dear authors,
I want to thank the authors for the changes realized. But, authors haven´t answered about my doubts in relation about the VT2 of the participants. We haven´t ensured that participants realized a HIIE if some of the have their VT2 to higher intensities at the intensity used (80% VO2max). I´m going to expose some minor comments, but, I want to say you that this paper don´t be accepted if authors don´t response to this question. Like physiologist, I cannot accept this paper if I´m not sure about it.
My minor comments are the next:
Include BMI and % body fat values. Actually it has been included that “An independent investigator not involved in any study procedures conducted the sequence randomization using the PC software RITA”. Nevertheless, I have revised the contributions section and I cannot revised the initial of this investigator for this assignment. Table 2: it´s necessary to include the p-value in a column. I propose a Figure 4.c with the RPE. I want to remember that it´s very important to ensure if all the participants realized an intensity higher to their VT2.Author Response
Reviewer 1:
Dear authors,
I want to thank the authors for the changes realized.
Thank you for the thorough review of our manuscript. We have carefully considered all of your comments and provided our responses below.
But, authors haven´t answered about my doubts in relation about the VT2 of the participants. We haven´t ensured that participants realized a HIIE if some of the have their VT2 to higher intensities at the intensity used (80% VO2max). I´m going to expose some minor comments, but, I want to say you that this paper don´t be accepted if authors don´t response to this question. Like physiologist, I cannot accept this paper if I´m not sure about it.
Thank you for once again raising your concerns about the exercise intensity used in our study. However, be believe this is based on a misunderstanding as pointed out in our previous reply. We clearly stated in our manuscript as well as our previous response letter that we did not calculated the intervention loads based on VO2max but instead based on peak power (i.e. maximal Wattage, Wmax). As pointed out earlier, this is a very crucial difference because in contrast to VO2max, peak power is much more robust because it is an actual performance parameter. Moreover, it is well known that VO2max and Wmax are not identical and depending on the protocol used for determination, relative loads based on Wmax are typically higher than those calculated based on VO2max. In the present study, VO2max (or actually VO2peak, as described in the manuscript) was only provided for descriptive purposes in order to characterize the sample investigated (similarly to body weight, height etc.).
Following our initial submission, you requested that we provide the second ventilatory thresholds of all subjects in order to proof that our loadings in fact were of high-intensity. Because we used a step test rather than a ramp protocol, ventilatory thresholds may not precisely be determined. This is why we provided the anaerobic threshold based on blood lactate concentrations and emphasized that the intensity used in our loading protocols was indeed well above this threshold (Load at anaerobic lactate threshold: 205 ± 25 W; 80% Wmax: 259 ± 24 W, i.e. on average 54 W above the anaerobic threshold). It is well established that the second ventilator thresholds corresponds accurately wth the anaerobic lactate thresholds (e.g. Ahmaidi et al., 1993; Pallarés et al. 2016; Elmer & Toney 2018). Moreover, we would like to stress that the blood lactate values assessed throughout the exercise loadings both in normoxia and hyperoxia ranged from 6 to 10 mmol·L-1 (see Figure 4a of our manuscript), which is clearly considered as high-intensity exercise based on previously determined intensity zones (e.g. Sylta, Tønnessen & Seiler, 2014).
Irrespective of the difference in loading determination, we believe that the question about the terminology (i.e. “high-intensity interval exercise” or “strenuous exercise”) does not in any way affect the interpretation of our results because the loads were kept equal in the normoxia and hyperoxia condition.
References:
Ahmaidi et al. 1993: https://link.springer.com/article/10.1007%2FBF00863396
Pallarés et al. 2016: https://journals.plos.org/plosone/article/file?id=10.1371/journal.pone.0163389&type=printable
Elmer & Toney, 2018: https://www.thieme-connect.com/products/ejournals/abstract/10.1055/s-0043-125448
Sylta, Tønnessen & Seiler, 2014: https://www.ncbi.nlm.nih.gov/pubmed/?term=TID+and+Seiler
My minor comments are the next:
Include BMI and % body fat values.
Thank you for this valuable advice. In light of your comments and those of reviewer 2, we have added information about the relative body fat and lean mass values. However, as reviewer 2 requested to delete BMI in the previous revision and considering that body weight and body height are already provided, we do not believe that BMI would still add to our reported findings.
Actually it has been included that “An independent investigator not involved in any study procedures conducted the sequence randomization using the PC software RITA”. Nevertheless, I have revised the contributions section and I cannot revised the initial of this investigator for this assignment.
You are right, this was a miswording and we thank you for pointing this out. Actually, an independent technician in house did the randomization, which we now clarified. However, according to authorship guidelines this does not justify authorship and, therefore, he was not included.
Table 2: it´s necessary to include the p-value in a column.
Thank you for this suggestion. However, we do believe, that adding another column including the p-values to table 2 leads to redundant information. All p-values but one were all smaller than p<0.001 which is indicated in the footer. The p-value for the single not statistical significant difference is described within the body of the text, similarly to that of all other comparisons.
I propose a Figure 4.c with the RPE. I want to remember that it´s very important to ensure if all the participants realized an intensity higher to their VT2.
We do agree and have added the Figure as requested. Please find the figure within the manuscript as Figure 4c.
Reviewer 2 Report
Thank you for addressing my concerns. The only issue I have remaining is the length of time for the VO2max tests. I think it should be mentioned in the manuscript, and addressed briefly. Since these tests are supposed to last 10-12 minutes and yours were nearly double that, they are likely a bit inaccurate. However, this doesn't greatly affect the results of your study.
Author Response
Reviewer 2:
"Thank you for addressing my concerns. The only issue I have remaining is the length of time for the VO2max tests. I think it should be mentioned in the manuscript, and addressed briefly. Since these tests are supposed to last 10-12 minutes and yours were nearly double that, they are likely a bit inaccurate. However, this doesn't greatly affect the results of your study."
Thank you for your careful review of our manuscript. This information has now been added.
Reviewer 3 Report
[*] In the Abstract and throughout, mean or percent change should be reported with 95% CI rather than SD. I think I only saw this in the Abstract, but please double check for other occurrences
[*] Introduction. It would be good to state a directed hypothesis toward your primary and secondary outcomes. In essence, what did you think would happen before performing the study.
Be honest. There is nothing wrong with rejecting your hypothesis should it be different than what you thought might happen should that have been the case.
[*] In the first paragraph of your Discussion please also accept your or reject your hypothesis.
[*] Please add a paragraph regarding Limitations and Generalisability of your findings.
[*] Did you register your trial?
Per Journal Instructions:
“Authors are strongly encouraged to pre-register clinical trials with an international clinical trials register or and to cite a reference to the registration in the Methods section. Suitable databases include clinicaltrials.gov, the EU Clinical Trials Register and those listed by the World Health Organisation International Clinical Trials Registry Platform.”
Admittedly this is loose verbiage, so if you did not it is not a deal breaker in my opinion, but you should make not of it should you wish to continue with the MDPI publications. Clinicialtrials.gov is free so no reason not to.
Author Response
Reviewer 3:
Thank you for your careful revision of our manuscript. Please find our responses below.
[*] In the Abstract and throughout, mean or percent change should be reported with 95% CI rather than SD. I think I only saw this in the Abstract, but please double check for other occurrences
Thank you for pointing this out. We have changed the abstract accordingly.
[*] Introduction. It would be good to state a directed hypothesis toward your primary and secondary outcomes. In essence, what did you think would happen before performing the study.
Be honest. There is nothing wrong with rejecting your hypothesis should it be different than what you thought might happen should that have been the case.
Thank you for this valuable comment. We have now clarified our hypotheses further.
[*] In the first paragraph of your Discussion please also accept your or reject your hypothesis.
This information has been added as suggested.
[*] Please add a paragraph regarding Limitations and Generalisability of your findings.
Thank you for pointing this out, we added the required information accordingly.
[*] Did you register your trial?
Per Journal Instructions:
“Authors are strongly encouraged to pre-register clinical trials with an international clinical trials register or and to cite a reference to the registration in the Methods section. Suitable databases include clinicaltrials.gov, the EU Clinical Trials Register and those listed by the World Health Organisation International Clinical Trials Registry Platform.”
Admittedly this is loose verbiage, so if you did not it is not a deal breaker in my opinion, but you should make not of it should you wish to continue with the MDPI publications. Clinicialtrials.gov is free so no reason not to.
As we mentioned earlier, this trial was part (amendment) of a larger clinical investigation with cancer patients, which is still ongoing and registered. The local ethics committee granted approval for this amendment; however, we did not change the study registry and thus there is no information about it in the registry. That is why we did not state any registration as not to cause any irritation. However, as you suggested we believe the preregistration may not be as crucial for this type of study as for purely clinical trials.
Round 2
Reviewer 1 Report
Dear autors,
It´s more important that authors believe the specification of the relative intensity of each participant. In this sense, authors said me that they employed the Wmax vs VO2max, but the VO2max determined in the test isn´t the real VO2max. For determining VO2max the test don´t exceed the 15 minutes and the protocol selected (increase 40 W each 3 minutes is so long). The VO2max registered is lower than real VO2max. In relation to second ventilatory threshold (VT2), authors must consider that lactate threshold is associated with first ventilatory threshold (VT1). Additionally, VT2 can be determined using step test. If authors want, they can use an algorithmic adjustment with VE value since the VE value at VT1 intensity (VT2 can be determined like second disproportionate increase during an incremental test).
From the perspective of Exercise Physiology the intensity of 80% Wmax can be a high intensity for some subjects, but not for all (depending of the intensity corresponding to VT2). In futire, I recommend to include an intensity a little higher (ie, 85% Wmax or 85-90% VO2max). But, in this moment I solicite the confirmation of the VT2 value for this participants. If authors don´t include this solicitation, I won´t accept the paper in the future.
Author Response
Thank you once again for presenting your point of view. We believe we have provided a wealth of evidence for a) the associations of the lactate threshold (i.e. second lactate threshold or termed OBLA as well in “older literature” anaerobic threshold) and VT2 as well as b) the classification of different loadings based on blood lactate concentrations and other means (e.g. RPE, HR). We apologize for the misunderstanding on the terminology used for describing lactate thresholds. Those presented to you in our previous response were that of the second lactate threshold and do, thus, correspond well to VT2.
Irrespective of the discussion on metabolic thresholds, we do not believe that the term “high-intensity interval exercise” does in any way affect the interpretation of our results because the absolute intensity was the same for a given subject in hyperoxia and normoxia.
This manuscript is a resubmission of an earlier submission. The following is a list of the peer review reports and author responses from that submission.
Round 1
Reviewer 1 Report
Dear authors,
I think that authors have realized an interesting research, novelty and with a high scientific soundness. Although I have some minor comments, I have a high concern about an aspect that I´m going to expose relative to discussion section.
My comments are the next:
Introduction:
It´s necessary to change the beginning of the manuscript. Before expressed the results of systematic reviews, it must be include a description of the mechanism and the effect of hyperoxia. Lines 71-72: it´s necessary to specify the variables that improve in the meta-analysis of Mallete et al. (2017).Materials and Methods:
Authors describe the participants like “recreationally physically active”, but the can be described based on a more specific description. For example, do the participants fulfill the ACSM´s exercise recommendations? It´s necessary to include the Ethics Committee´s code. Did the experimental session realized at the same time slot? Line 98: it´s necessary to specify the percentage.Statistical analysis:
Differences between the conditions in the basic blood count values as well as for the ratio of SS1/2 to EImax must be determined using a repeated measure ANOVA and not a one-way ANOVA. Authors must considerer that they are analyzing different values of a same subjects and one-way ANOVA is employed when we want to compare different groups.Results:
Authors must include the p-value of ANOVA in each variable and for the two factor (time and experimental condition) in the two-way repeated measure ANOVA.Discussion:
I have an important doubt about this section. Thereby, authors indicate that it was realized a “high-intensity interval exercise”. But nevertheless, I have some doubts about it. First of all, the volume of the interval session is very low for physical active males only 5 bouts of 3 minutes. Additionally, the intensity was only 80% VO2max, generally the intensities used for athlethes when realized bouts of 3 minutes (1,000 m in trained runners, approximately) is in a range of 85-95% VO2max. That is consequence that trained athlethes have their second ventilatory threshold to an intensity near to 85% VO2max. In this sample, I think that it´s possible that VT2 it´s located upper 80% VO2max. In this hypothetical situation, the study would have analyzed a moderate intensity (not a high intensity). Due the authors have the data of gas exchange, I propose to determine the VT2 for cheking if the subjects used a moderate or high intensity. In the chase that some participantes realized the experimental sessions at moderate or high intensity, I propose to realize a new analysis with two different group with different intensities (moderate and high intensity).Reviewer 2 Report
I have provided some specific comments below.
Abstract:
HIIE was never defined – please do so.
No comma necessary after the word “age” in line 24.
Line 26 – shear stress of what?
RBC-D hasn’t been defined yet either.
Introduction:
The introduction should, in my opinion, have some background on the limits of oxygen consumption. There are plenty of good articles (and reviews) on the topic and this is paramount to this study. Why would increasing the amount of inhaled oxygen improve performance?
Line 49-50 – needs revised as it’s not grammatically correct.
Line 70 – “performance” is vague – what kind of performance test was it?
Line 76 – why a flow rate of only 4 L/min? I think some rationale for this would be helpful.
Hypothesis? And what does “superior physiological function” actually mean?
Methods:
Table 1 – Don’t need height/mass if you have BMI. I’d take out the former two and keep the latter.
Line 112 – What was the average test time of the VO2max test?
Line 119 – Some studies have shown that using a 15-breath rolling average, and the peak value obtained from that method, is more valid. I suggest looking into this analysis method.
Line 119 – Interesting method for calculating Wmax. Is there a reference for this?
Line 122 – Why this protocol for testing performance? Seems like another VO2max test or a TTE would be good to use as well. A 3-min exercise bout is heavily O2 dependent, but not as much as a TTE would be.
Results:
Line 228 – Why are the blood lactate values presented as percentages? Why not mmol/L?
Throughout the results section, you could cut out a lot of unnecessary words to make the section cleaner. For example – instead of “No statistically significant between-condition differences were observed…” you could use “There was no difference between conditions…” or “No between-condition differences were…”
Line 240 – Don’t need to define RPE here, but should define it back in the methods section. Also, it’s “rating of perceived exertion”, not “rated perceived exertion”.
Discussion:
Referring back to the comments about the intro section… I think this intro/discussion should be formed around the limits of O2 uptake in humans, which it is currently not. Although I think the exercise test chosen was inappropriate as well so it may be a moot point. It was mentioned in the intro that the mechanism behind the improved TTE/VO2max improvement may be due to maintained muscle O2 – so why didn’t you just recreate those studies and look more into the mechanism? Why choose a different exercise protocol??
Reviewer 3 Report
Hyperoxia
Overall, this is a well written paper that flows well and reads easily. Unfortunately, it does not conform to CONSORT reporting standards which is a requirement of this journal. All of these criteria carry a requirement of pre-specification. If provided the authors would do this after the fact, which is inappropriate in many instances.
“CONSORT Statement
Sports requires a completed CONSORT 2010 checklist and flow diagram as a condition of submission when reporting the results of a randomized trial. Templates for these can be found here or on the CONSORT website (http://www.consort-statement.org) which also describes several CONSORT checklist extensions for different designs and types of data beyond two group parallel trials. At minimum, your article should report the content addressed by each item of the checklist. Meeting these basic reporting requirements will greatly improve the value of your trial report and may enhance its chances for eventual publication.”
For example:
[1] Did you have a hypothesis or was this an exploratory study? If you did have a hypothesis it would be nice of stated it. If not, you should consider labelling this an exploratory study.
[2] Sports uses a CONSORT based reporting system, which asks authors to provide a check list of items.
6a requires “Outcomes 6a Completely defined pre-specified primary and secondary outcome measures, including how and when they were assessed.”
I don’t see an effort for prioritization of outcomes in this paper and since the statement asks for pre-specified outcomes.
[3] Line 7a of the CONSORT requires verbiage of how the sample size was determined. I do not see this either.
[4] Lines 9 and 10 speak to allocation procedures. This is not reported.
[5] Line 90. Ethical approval. What institution provided and what number was assigned to the review approval.
[6] Line 189. ANOVA procedures should be reported with partial eta squared, not Cohen’s D.
http://imaging.mrc-cbu.cam.ac.uk/statswiki/FAQ/effectSize?fbclid=IwAR3pZl_6nCEQ9HZEvJrub8HP6_-z4waoHU6tW8P2BKxpV_He3sVsBj7f6PU
[7] Percent or mean change should be reported with 95% CI. See line 17a of the CONSORT check list you were asked to complete.
[8] Consort Check List, “Limitations 20 Trial limitations, addressing sources of potential bias, imprecision, and, if relevant, multiplicity of analyses.”
This was not provided.
[9] “Generalisability 21 Generalisability (external validity, applicability) of the trial findings.
This was not provided.